# Iterative Amortized Policy Optimization

**Joseph Marino**[*]
California Institute of Technology

**Alexandre Piché**
Mila, Université de Montréal

**Alessandro Davide Ialongo**
University of Cambridge

**Yisong Yue**
California Institute of Technology

## Abstract

Policy networks are a central feature of deep reinforcement learning (RL) algorithms for continuous control, enabling the estimation and sampling of high-value actions. From the variational inference perspective on RL, policy networks, when used with entropy or KL regularization, are a form of *amortized optimization*, optimizing network parameters rather than the policy distributions directly. However, *direct* amortized mappings can yield suboptimal policy estimates and restricted distributions, limiting performance and exploration. Given this perspective, we consider the more flexible class of *iterative* amortized optimizers. We demonstrate that the resulting technique, iterative amortized policy optimization, yields performance improvements over direct amortization on benchmark continuous control tasks. Accompanying code: `github.com/joelouismarino/variational_rl`.

## 1 Introduction

Reinforcement learning (RL) algorithms involve policy evaluation and policy optimization [73]. Given a policy, one can estimate the value for each state or state-action pair following that policy, and given a value estimate, one can improve the policy to maximize the value. This latter procedure, policy optimization, can be challenging in continuous control due to instability and poor asymptotic performance. In deep RL, where policies over continuous actions are often parameterized by deep networks, such issues are typically tackled using regularization from previous policies [67, 68] or by maximizing policy entropy [57, 23]. These techniques can be interpreted as variational inference [51], using optimization to infer a policy that yields high expected return while satisfying prior policy constraints. This smooths the optimization landscape, improving stability and performance [3].

However, one subtlety arises: when used with entropy or KL regularization, policy networks perform *amortized* optimization [26]. That is, rather than optimizing the action distribution, e.g., mean and variance, many deep RL algorithms, such as soft actor-critic (SAC) [31, 32], instead optimize a network to output these parameters, *learning* to optimize the policy. Typically, this is implemented as a direct mapping from states to action distribution parameters. While such *direct* amortization schemes have improved the efficiency of variational inference as "encoder" networks [44, 64, 56], they also suffer from several drawbacks: *1)* they tend to provide suboptimal estimates [20, 43, 55], yielding a so-called "amortization gap" in performance [20], *2)* they are restricted to a single estimate [27], thereby limiting exploration, and *3)* they cannot generalize to new objectives, unlike, e.g., gradient-based [36] or gradient-free optimizers [66].

Inspired by techniques and improvements from variational inference, we investigate *iterative* amortized policy optimization. Iterative amortization [55] uses gradients or errors to iteratively update the parameters of a distribution. Unlike direct amortization, which receives gradients only *after*

---

[*]Now at DeepMind, London, UK. Correspondence to `josephmarino@deepmind.com`.

35th Conference on Neural Information Processing Systems (NeurIPS 2021).

outputting the distribution, iterative amortization uses these gradients *online*, thereby learning to iteratively optimize. In generative modeling settings, iterative amortization empirically outperforms direct amortization [55, 54] and can find multiple modes of the optimization landscape [27].

The contributions of this paper are as follows:

- We propose iterative amortized policy optimization, exploiting a new, fruitful connection between amortized variational inference and policy optimization.
- Using the suite of MuJoCo environments [78, 12], we demonstrate performance improvements over direct amortized policies, as well as more complex flow-based policies.
- We demonstrate novel benefits of this amortization technique: improved accuracy, providing multiple policy estimates, and generalizing to new objectives.

## 2 Background

### 2.1 Preliminaries

We consider Markov decision processes (MDPs), where $\mathbf{s}_t \in \mathcal{S}$ and $\mathbf{a}_t \in \mathcal{A}$ are the state and action at time $t$, resulting in reward $r_t = r(\mathbf{s}_t, \mathbf{a}_t)$. Environment state transitions are given by $\mathbf{s}_{t+1} \sim p_{\text{env}}(\mathbf{s}_{t+1}|\mathbf{s}_t, \mathbf{a}_t)$, and the agent is defined by a parametric distribution, $p_\theta(\mathbf{a}_t|\mathbf{s}_t)$, with parameters $\theta$.[2] The discounted sum of rewards is denoted as $\mathcal{R}(\tau) = \sum_t \gamma^t r_t$, where $\gamma \in (0, 1]$ is the discount factor, and $\tau = (\mathbf{s}_1, \mathbf{a}_1, \dots)$ is a trajectory. The distribution over trajectories is:

$$p(\tau) = \rho(\mathbf{s}_1) \prod_{t=1}^{T-1} p_{\text{env}}(\mathbf{s}_{t+1}|\mathbf{s}_t, \mathbf{a}_t)p_\theta(\mathbf{a}_t|\mathbf{s}_t), \tag{1}$$

where the initial state is drawn from the distribution $\rho(\mathbf{s}_1)$. The standard RL objective consists of maximizing the expected discounted return, $\mathbb{E}_{p(\tau)}[\mathcal{R}(\tau)]$. For convenience of presentation, we use the undiscounted setting ($\gamma = 1$), though the formulation can be applied with any valid $\gamma$.

### 2.2 KL-Regularized Reinforcement Learning

Various works have formulated RL, planning, and control problems in terms of probabilistic inference [21, 8, 79, 77, 11, 51]. These approaches consider the agent-environment interaction as a graphical model, then convert reward maximization into maximum marginal likelihood estimation, learning and inferring a policy that results in maximal reward. This conversion is accomplished by introducing one or more binary observed variables [19], denoted as $\mathcal{O}$ for "optimality" [51], with

$$p(\mathcal{O} = 1|\tau) \propto \exp\big(\mathcal{R}(\tau)/\alpha\big),$$

where $\alpha$ is a temperature hyper-parameter. We would like to infer latent variables, $\tau$, and learn parameters, $\theta$, that yield the maximum log-likelihood of optimality, i.e., $\log p(\mathcal{O} = 1)$. Evaluating this likelihood requires marginalizing the joint distribution, $p(\mathcal{O} = 1) = \int p(\tau, \mathcal{O} = 1)d\tau$. This involves averaging over all trajectories, which is intractable in high-dimensional spaces. Instead, we can use variational inference to lower bound this objective, introducing a structured approximate posterior distribution:

$$\pi(\tau|\mathcal{O}) = \rho(\mathbf{s}_1) \prod_{t=1}^{T-1} p_{\text{env}}(\mathbf{s}_{t+1}|\mathbf{s}_t, \mathbf{a}_t)\pi(\mathbf{a}_t|\mathbf{s}_t, \mathcal{O}). \tag{2}$$

This provides the following lower bound on the objective:

$$\log p(\mathcal{O} = 1) = \log \int p(\mathcal{O} = 1|\tau)p(\tau)d\tau \tag{3}$$

$$\geq \int \pi(\tau|\mathcal{O})\left[\log \frac{p(\mathcal{O} = 1|\tau)p(\tau)}{\pi(\tau|\mathcal{O})}\right]d\tau \tag{4}$$

$$= \mathbb{E}_\pi[\mathcal{R}(\tau)/\alpha] - D_{\text{KL}}(\pi(\tau|\mathcal{O})\|p(\tau)). \tag{5}$$

---

[2]In this paper, we consider the entropy-regularized case, where $p_\theta(\mathbf{a}_t|\mathbf{s}_t) = \mathcal{U}(-1, 1)$, i.e., uniform. However, we present the derivation for the KL-regularized case for full generality.

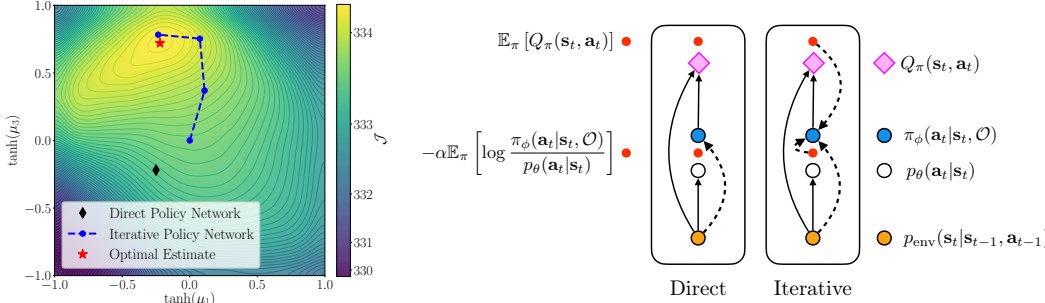

Figure 1: **Amortization**. **Left**: Optimization over two dimension of the policy mean, $\mu_1$ and $\mu_3$, for a particular state. A direct amortized policy network outputs a suboptimal estimate, yielding an *amortization gap* in performance. An iterative amortized policy network finds an improved estimate. **Right**: Diagrams of direct and iterative amortization. Larger circles denote distributions, and smaller red circles denote terms in the objective, $\mathcal{J}$ (Eq. 8). Dashed arrows denote amortization. Iterative amortization uses gradient feedback during optimization, while direct amortization does not.

Equivalently, we can multiply by $\alpha$, defining the variational RL objective as:

$$\mathcal{J}(\pi, \theta) \equiv \mathbb{E}_\pi[\mathcal{R}(\tau)] - \alpha D_{\mathrm{KL}}(\pi(\tau|\mathcal{O})\|p(\tau)). \tag{6}$$

This objective consists of the expected return (i.e., the standard RL objective) and a KL divergence between $\pi(\tau|\mathcal{O})$ and $p(\tau)$. In terms of states and actions, this objective is:

$$\mathcal{J}(\pi, \theta) = \mathbb{E}_{\substack{\mathbf{s}_t, r_t \sim p_{\mathrm{env}} \\ \mathbf{a}_t \sim \pi}} \left[ \sum_{t=1}^{T} r_t - \alpha \log \frac{\pi(\mathbf{a}_t|\mathbf{s}_t, \mathcal{O})}{p_\theta(\mathbf{a}_t|\mathbf{s}_t)} \right]. \tag{7}$$

At a given timestep, $t$, one can optimize this objective by estimating the future terms in the sum using a "soft" action-value ($Q_\pi$) network [30] or model [62]. For instance, sampling $\mathbf{s}_t \sim p_{\mathrm{env}}$, slightly abusing notation, we can write the objective at time $t$ as:

$$\mathcal{J}(\pi, \theta) = \mathbb{E}_\pi\left[Q_\pi(\mathbf{s}_t, \mathbf{a}_t)\right] - \alpha D_{\mathrm{KL}}(\pi(\mathbf{a}_t|\mathbf{s}_t, \mathcal{O})\|p_\theta(\mathbf{a}_t|\mathbf{s}_t)). \tag{8}$$

Policy optimization in the KL-regularized setting corresponds to maximizing $\mathcal{J}$ w.r.t. $\pi$. We often consider parametric policies, in which $\pi$ is defined by distribution parameters, $\boldsymbol{\lambda}$, e.g., Gaussian mean, $\boldsymbol{\mu}$, and variance, $\boldsymbol{\sigma}^2$. In this case, policy optimization corresponds to maximizing:

$$\boldsymbol{\lambda} \leftarrow \arg\max_{\boldsymbol{\lambda}} \mathcal{J}(\pi, \theta). \tag{9}$$

Optionally, we can then also learn the policy prior parameters, $\theta$ [1].

## 2.3 Entropy & KL Regularized Policy Networks Perform Direct Amortization

Policy-based approaches to RL typically do not directly optimize the action distribution parameters, e.g., through gradient-based optimization. Instead, the action distribution parameters are output by a function approximator (deep network), $f_\phi$, which is trained using deterministic [70, 52] or stochastic gradients [83, 35]. When combined with entropy or KL regularization, this policy network is a form of *amortized* optimization [26], learning to estimate policies. Again, denoting the action distribution parameters, e.g., mean and variance, as $\boldsymbol{\lambda}$, for a given state, $\mathbf{s}$, we can express this direct mapping as

$$\boldsymbol{\lambda} \leftarrow f_\phi(\mathbf{s}), \qquad \text{(direct amortization)} \tag{10}$$

denoting the corresponding policy as $\pi_\phi(\mathbf{a}|\mathbf{s}, \mathcal{O}; \boldsymbol{\lambda})$. Thus, $f_\phi$ attempts to *learn* to optimize Eq. 9. This setup is shown in Figure 1 (Right). Without entropy or KL regularization, i.e. $\pi_\phi(\mathbf{a}|\mathbf{s}) = p_\theta(\mathbf{a}|\mathbf{s})$, we can instead interpret the network as directly integrating the LHS of Eq. 4, which is less efficient and more challenging. Regularization smooths the optimization landscape, yielding more stable improvement and higher asymptotic performance [3].

Viewing policy networks as a form of direct amortized variational optimizer (Eq. 10) allows us to see that they are similar to "encoder" networks in variational autoencoders (VAEs) [44, 64]. However, there are several drawbacks to direct amortization.

**Algorithm 1** Direct Amortization

Initialize $\phi$
**for** each environment step **do**
    $\boldsymbol{\lambda} \leftarrow f_\phi(\mathbf{s}_t)$
    $\mathbf{a}_t \sim \pi_\phi(\mathbf{a}_t | \mathbf{s}_t, \mathcal{O}; \boldsymbol{\lambda})$
    $\mathbf{s}_{t+1} \sim p_{\text{env}}(\mathbf{s}_{t+1} | \mathbf{s}_t, \mathbf{a}_t)$
**end for**
**for** each training step **do**
    $\phi \leftarrow \phi + \eta \nabla_\phi \mathcal{J}$
**end for**

**Algorithm 2** Iterative Amortization

Initialize $\phi$
**for** each environment step **do**
    Initialize $\boldsymbol{\lambda}$
    **for** each policy optimization iteration **do**
        $\boldsymbol{\lambda} \leftarrow f_\phi(\mathbf{s}_t, \boldsymbol{\lambda}, \nabla_{\boldsymbol{\lambda}} \mathcal{J})$
    **end for**
    $\mathbf{a}_t \sim \pi_\phi(\mathbf{a}_t | \mathbf{s}_t, \mathcal{O}; \boldsymbol{\lambda})$
    $\mathbf{s}_{t+1} \sim p_{\text{env}}(\mathbf{s}_{t+1} | \mathbf{s}_t, \mathbf{a}_t)$
**end for**
**for** each training step **do**
    $\phi \leftarrow \phi + \eta \nabla_\phi \mathcal{J}$
**end for**

**Amortization Gap.** Direct amortization results in suboptimal approximate posterior estimates, with the resulting gap in the variational bound referred to as the *amortization gap* [20]. Thus, in the RL setting, an amortized policy, $\pi_\phi$, results in worse performance than the optimal policy within the parametric policy class, denoted as $\widehat{\pi}$. The amortization gap is the gap in following inequality:

$$\mathcal{J}(\pi_\phi, \theta) \leq \mathcal{J}(\widehat{\pi}, \theta).$$

Because $\mathcal{J}$ is a variational bound on the RL objective, i.e., expected return, a looser bound, due to amortization, prevents one from more completely optimizing this objective.

This is shown in Figure 1 (Left),[3] where $\mathcal{J}$ is plotted over two dimensions of the policy mean at a particular state in the MuJoCo environment `Hopper-v2`. The estimate of a direct amortized policy (◆) is suboptimal, far from the optimal estimate (★). While the relative difference in the objective is relatively small, suboptimal estimates prevent sampling and exploring high-value regions of the action-space. That is, suboptimal estimates have only a *minor* impact on evaluation performance (see Appendix B.6) but hinder effective data collection.

**Single Estimate.** Direct amortization is limited to a single, static estimate. In other words, if there are multiple high-value regions of the action-space, a uni-modal (e.g., Gaussian) direct amortized policy is restricted to only one region, thereby limiting exploration. Note that this is an additional restriction beyond simply considering uni-modal distributions, as a generic optimization procedure may arrive at multiple uni-modal estimates depending on initialization and stochastic sampling (see Section 3.2). While multi-modal distributions reduce the severity of this restriction [74, 29], the other limitations of direct amortization still persist.

**Inability to Generalize Across Objectives.** Direct amortization is a feedforward procedure, receiving gradients from the objective only *after* estimation. This is contrast to other forms of optimization, which receive gradients (feedback) *during* estimation. Thus, unlike other optimizers, direct amortization is incapable of generalizing to new objectives, e.g., if $Q_\pi(\mathbf{s}, \mathbf{a})$ or $p_\theta(\mathbf{a}|\mathbf{s})$ change, which is a desirable capability for adapting to new tasks or environments.

To improve upon this scheme and overcome these drawbacks, in Section 3, we turn to a technique developed in generative modeling, *iterative amortization* [55], retaining the efficiency of amortization while employing a more flexible iterative estimation procedure.

## 2.4 Related Work

Previous works have investigated methods for improving policy optimization. QT-Opt [41] uses the cross-entropy method (CEM) [66], an iterative derivative-free optimizer, to optimize a $Q$-value estimator for robotic grasping. CEM and related methods are also used in model-based RL for performing model-predictive control [60, 14, 62, 33]. Gradient-based policy optimization [36, 71, 10], in contrast, is less common, however, gradient-based optimization can also be combined with CEM

---

[3] Additional 2D plots are shown in Figure 19 in the Appendix.

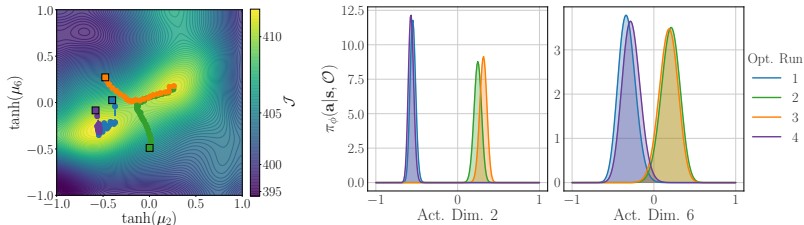

Figure 2: **Estimating Multiple Policy Modes**. Unlike direct amortization, which is restricted to a single estimate, iterative amortization can effectively sample from multiple high-value action modes. This is shown for a particular state in `Ant-v2`, showing multiple optimization runs across two action dimensions (**Left**). Each square denotes an initialization. The optimizer finds both modes, with the densities plotted on the **Right**. This capability provides increased flexibility in action exploration.

[5]. Most policy-based methods use direct amortization, either using a feedforward [31] or recurrent [28] network. Similar approaches have also been applied to model-based value estimates [13, 16, 4], as well as combining direct amortization with model predictive control [50] and planning [65]. A separate line of work has explored improving the policy distribution, using normalizing flows [29, 74] and latent variables [76]. In principle, iterative amortization can perform policy optimization in each of these settings.

Iterative amortized policy optimization is conceptually similar to negative feedback control [7], using errors to update policy estimates. However, while conventional feedback control methods are often restricted in their applicability, e.g., linear systems and quadratic cost, iterative amortization is generally applicable to any differentiable control objective. This is analogous to the generalization of Kalman filtering [42] to amortized filtering [54] for state estimation.

# 3 Iterative Amortized Policy Optimization

## 3.1 Formulation

Iterative amortization [55] utilizes errors or gradients to update the approximate posterior distribution parameters. While various forms exist, we consider gradient-encoding models [6] due to their generality. Compared with direct amortization (Eq. 10), we use iterative amortized optimizers of the general form

$$\boldsymbol{\lambda} \leftarrow f_\phi(\mathbf{s}, \boldsymbol{\lambda}, \nabla_{\boldsymbol{\lambda}} \mathcal{J}), \qquad \text{(iterative amortization)} \qquad (11)$$

also shown in Figure 1 (Right), where $f_\phi$ is a deep network and $\boldsymbol{\lambda}$ are the action distribution parameters. For example, if $\pi = \mathcal{N}(\mathbf{a}; \boldsymbol{\mu}, \text{diag}(\boldsymbol{\sigma}^2))$, then $\boldsymbol{\lambda} \equiv [\boldsymbol{\mu}, \boldsymbol{\sigma}]$. Technically, $\mathbf{s}$ is redundant, as the state dependence is already captured in $\mathcal{J}$, but this can empirically improve performance [55]. In practice, the update is carried out using a "highway" gating operation [38, 72]. Denoting $\boldsymbol{\omega}_\phi \in [0, 1]$ as the gate and $\boldsymbol{\delta}_\phi$ as the update, both of which are output by $f_\phi$, the gating operation is expressed as

$$\boldsymbol{\lambda} \leftarrow \boldsymbol{\omega}_\phi \odot \boldsymbol{\lambda} + (\mathbf{1} - \boldsymbol{\omega}_\phi) \odot \boldsymbol{\delta}_\phi, \qquad (12)$$

where $\odot$ denotes element-wise multiplication. This update is typically run for a fixed number of steps, and, as with a direct policy, the iterative optimizer is trained using stochastic gradient estimates of $\nabla_\phi \mathcal{J}$, obtained through the path-wise derivative estimator [44, 64, 35]. Because the gradients $\nabla_{\boldsymbol{\lambda}} \mathcal{J}$ must be estimated online, i.e., during policy optimization, this scheme requires some way of estimating $\mathcal{J}$ online through a parameterized $Q$-value network [58] or a differentiable model [35].

## 3.2 Benefits of Iterative Amortization

**Reduced Amortization Gap.** Iterative amortized optimizers are more flexible than their direct counterparts, incorporating feedback from the objective *during* policy optimization (Algorithm 2), rather than only *after* optimization (Algorithm 1). Increased flexibility improves the accuracy of optimization, thereby tightening the variational bound [55, 54]. We see this flexibility in Figure 1 (Left), where an iterative amortized policy network iteratively refines the policy estimate (●), quickly arriving near the optimal estimate.

**Multiple Estimates.** Iterative amortization, by using stochastic gradients and random initialization, can traverse the optimization landscape. As with any iterative optimization scheme, this allows iterative amortization to obtain multiple valid estimates, referred to as "multi-stability" in the generative modeling literature [27]. We illustrate this capability across two action dimensions in Figure 2 for a state in the `Ant-v2` MuJoCo environment. Over multiple policy optimization runs, iterative amortization finds multiple modes, sampling from two high-value regions of the action space. This provides increased flexibility in action exploration, despite only using a uni-modal policy distribution.

**Generalization Across Objectives.** Iterative amortization uses the gradients of the objective *during* optimization, i.e., feedback, allowing it to potentially generalize to new or updated objectives. We see this in Figure 1 (Left), where iterative amortization, despite being trained with a *different* value estimator, is capable of generalizing to this new objective. We demonstrate this capability further in Section 4. This opens the possibility of accurately and efficiently performing policy optimization in new settings, e.g., a rapidly changing model or new tasks.

### 3.3   Consideration: Mitigating Value Overestimation

Why are more powerful policy optimizers typically not used in practice? Part of the issue stems from value overestimation. Model-free approaches generally estimate $Q_\pi$ using function approximation and temporal difference learning. However, this has the pitfall of value overestimation, i.e., positive bias in the estimate, $\widehat{Q}_\pi$ [75]. This issue is tied to uncertainty in the value estimate, though it is distinct from optimism under uncertainty. If the policy can exploit regions of high uncertainty, the resulting target values will introduce positive bias into the estimate. More flexible policy optimizers exacerbate the problem, exploiting this uncertainty to a greater degree. Further, a rapidly changing policy increases the difficulty of value estimation [63].

Various techniques have been proposed for mitigating value overestimation in deep RL. The most prominent technique, double deep $Q$-network [81] maintains two $Q$-value estimates [80], attempting to decouple policy optimization from value estimation. Fujimoto et al. [25] apply and improve upon this technique for actor-critic settings, estimating the target $Q$-value as the minimum of two $Q$-networks, $Q_{\psi_1}$ and $Q_{\psi_2}$:

$$\widehat{Q}_\pi(\mathbf{s}, \mathbf{a}) = \min_{i=1,2} Q_{\psi'_i}(\mathbf{s}, \mathbf{a}),$$

where $\psi'_i$ denotes the target network parameters. As noted by Fujimoto et al. [25], this not only counteracts value overestimation, but also penalizes high-variance value estimates, because the minimum decreases with the variance of the estimate. Ciosek et al. [15] noted that, for a bootstrapped ensemble of two $Q$-networks, the minimum operation can be interpreted as estimating

$$\widehat{Q}_\pi(\mathbf{s}, \mathbf{a}) = \mu_Q(\mathbf{s}, \mathbf{a}) - \beta \sigma_Q(\mathbf{s}, \mathbf{a}), \tag{13}$$

with mean $\mu_Q(\mathbf{s}, \mathbf{a}) \equiv \frac{1}{2} \sum_{i=1,2} Q_{\psi'_i}(\mathbf{s}, \mathbf{a})$, standard deviation $\sigma_Q(\mathbf{s}, \mathbf{a}) \equiv (\frac{1}{2} \sum_{i=1,2}(Q_{\psi'_i}(\mathbf{s}, \mathbf{a}) - \mu_Q(\mathbf{s}, \mathbf{a}))^2)^{1/2}$, and $\beta = 1$. Thus, to further penalize high-variance value estimates, preventing value overestimation, we can increase $\beta$. For large $\beta$, however, value estimates become overly pessimistic, negatively impacting training. Thus, $\beta$ reduces target value variance at the cost of increased bias.

Due to the flexibility of iterative amortization, the default $\beta = 1$ results in increased value bias and a more rapidly changing policy as compared with direct amortization (Figure 3). Further penalizing high-variance target values ($\beta = 2.5$) reduces value overestimation and improves stability. For details, see Appendix A.2. Recent techniques for mitigating overestimation have been proposed, such as adjusting $\alpha$ [22]. In offline RL, this issue has been tackled through the action prior [24, 48, 84] or by altering $Q$-network training [2, 49]. While such techniques could be used here, increasing $\beta$ provides a simple solution with no additional computational overhead. This is a meaningful insight toward applying more powerful policy optimizers.

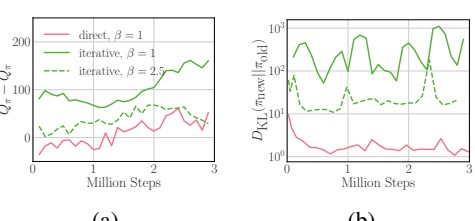

(a)          (b)

Figure 3: **Mitigating Value Overestimation**. With $\beta = 1$, iterative amortization results in (**a**) higher value overestimation and (**b**) a more rapidly changing policy as compared with direct amortization. Increasing $\beta$ helps to mitigate these issues.

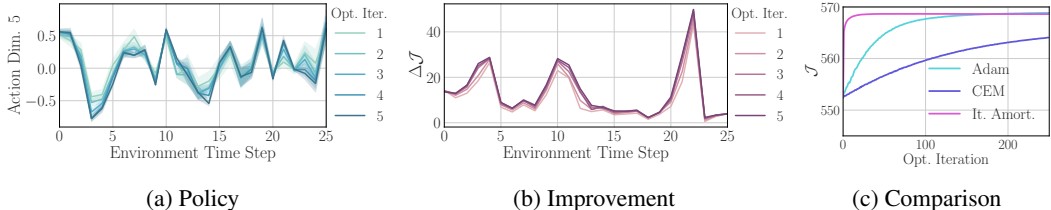

(a) Policy       (b) Improvement       (c) Comparison

Figure 4: **Policy Optimization**. Visualization over time steps of **(a)** one dimension of the policy distribution and **(b)** the improvement in the objective, $\Delta \mathcal{J}$, across policy optimization iterations. **(c)** Comparison of iterative amortization with Adam [45] (gradient-based) and CEM [66] (gradient-free). Iterative amortization is an order of magnitude more efficient.

## 4 Experiments

### 4.1 Setup

To focus on policy optimization, we implement iterative amortized policy optimization using the soft actor-critic (SAC) setup described by Haarnoja et al. [32]. This uses two $Q$-networks, uniform action prior, $p_\theta(\mathbf{a}|\mathbf{s}) = \mathcal{U}(-1, 1)$, and a tuning scheme for the temperature, $\alpha$. In our experiments, "direct" refers to direct amortization employed in SAC, i.e., a direct policy network, and "iterative" refers to iterative amortization. Both approaches use the *same* network architecture, adjusting only the number of inputs and outputs to accommodate gradients, current policy estimates, and gated updates (Sec. 3.1). Unless otherwise stated, we use 5 iterations per time step for iterative amortization, following [55]. For details, refer to Appendix A and Haarnoja et al. [31, 32].

### 4.2 Analysis

#### 4.2.1 Visualizing Policy Optimization

We provide 2D visualizations of iterative amortized policy optimization in Figures 1 & 2, with additional 2D plots in Appendix B.5. In Figure 4, we visualize iterative refinement using a single action dimension from Ant-v2 across time steps. The refinements in Figure 4a give rise to the objective improvements in Figure 4b. We compare with Adam [45] (gradient-based) and CEM [66] (gradient-free) in Figure 4c, where iterative amortization is *an order of magnitude* more efficient. This trend is consistent across environments, as shown in Appendix B.4.

#### 4.2.2 Performance Comparison

We evaluate iterative amortized policy optimization on the suite of MuJoCo [78] continuous control tasks from OpenAI gym [12]. In Figure 5, we compare the cumulative reward of direct and iterative amortized policy optimization across environments. Each curve shows the mean and $\pm$ standard deviation of 5 random seeds. In all cases, iterative amortized policy optimization matches or outperforms the baseline direct amortized method, both in sample efficiency and final performance. Iterative amortization also yields more consistent, lower variance performance.

#### 4.2.3 Improved Exploration: Multiple Policy Modes

As described in Section 3.2, iterative amortization is capable of obtaining multiple estimates, i.e., multiple modes of the optimization objective. To confirm that iterative amortization has captured multiple modes, at the end of training, we take an iterative agent trained on Walker2d-v2 and histogram the distances between policy means across separate runs of policy optimization per state (Fig. 7a). For the state with the largest distance, we plot 2D projections of the optimization objective, $\mathcal{J}$, across action dimensions in Figure 7b, as well as the policy density across 10 optimization runs (Fig. 7c). The multi-modal policy optimization surface shown in Figure 7b results in the multi-modal policy in Figure 7c. Additional results on other environments are presented in Appendix B.7.

To better understand whether the performance benefits of iterative amortization are coming purely from improved exploration via multiple modes, we also compare with direct amortization with a

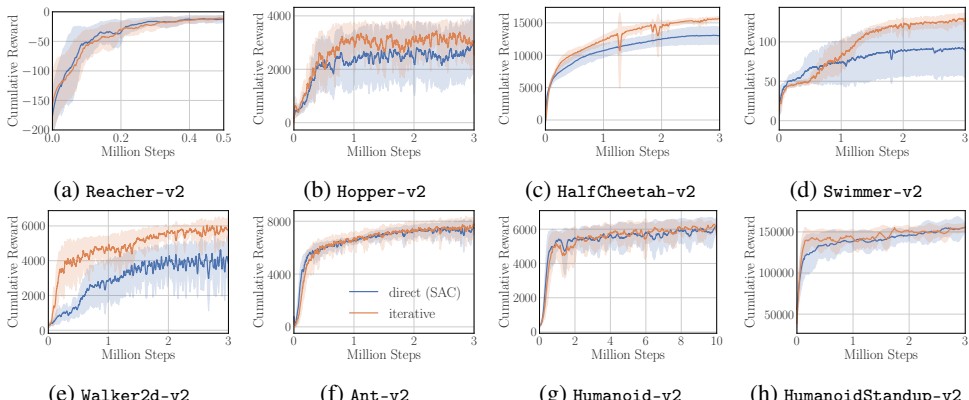

Figure 5: **Performance Comparison**. Iterative amortized policy optimization performs comparably with or better than direct amortization across MuJoCo environments. Curves show the mean ± std. dev. of performance over 5 random seeds.

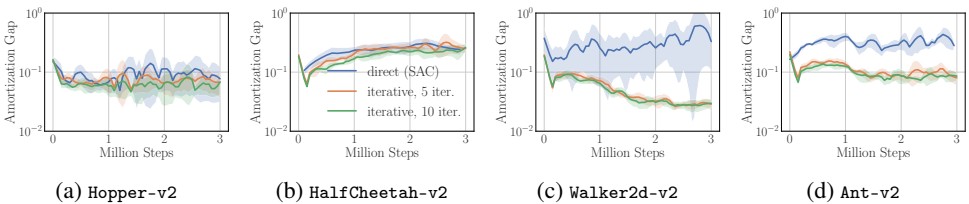

Figure 6: **Amortization Gap**. Iterative amortization achieves similar or lower amortization gaps than direct amortization. Gaps are estimated using stochastic gradient-based optimization over 100 random states. Curves show the mean and ± std. dev. over 5 random seeds.

multi-modal policy distribution. This is formed using inverse autoregressive flows [46], a type of normalizing flow (NF). Results are presented in Appendix B.2. Using a multi-modal policy reduces the performance deficiencies on `Hopper-v2` and `Walker2d-v2`, indicating that much of the benefit of iterative amortization is due to lifting direct amortization's restriction to a single, uni-modal policy estimate. Yet, direct + NF still struggles on `HalfCheetah-v2` compared with iterative amortization, suggesting that more complex, multi-modal distributions are not the *only* consideration.

### 4.2.4 Improved Optimization: Amortization Gap

To evaluate policy optimization accuracy, we estimate per-step amortization gaps, performing additional iterations of gradient ascent on $\mathcal{J}$ w.r.t. the policy parameters, $\lambda \equiv [\mu, \sigma]$ (see Appendix A.3). To analyze generalization, we also evaluate the iterative agents trained with 5 iterations for an additional 5 amortized iterations. Results are shown in Figure 6. We emphasize that it is challenging to *directly* compare amortization gaps across optimization schemes, as these involve different value functions, and therefore different objectives. Likewise, we estimate the amortization gap using the learned $Q$-networks, which may be biased (Figure 3). Nevertheless, we find that iterative amortized policy optimization achieves, on average, lower amortization gaps than direct amortization across all environments. Additional amortized iterations at evaluation yield further estimated improvement, demonstrating generalization beyond the optimization horizon used during training.

The amortization gaps are small relative to the objective, playing a negligible role in *evaluation* performance (see Appendix B.6). Rather, improved policy optimization is helpful for *training*, allowing the agent to explore states where value estimates are highest. To probe this further, we train iterative amortized policy optimization while varying the number of iterations per step in $\{1, 2, 5\}$, yielding optimizers with varying degrees of accuracy. Note that each optimizer is, in theory, capable of finding multiple modes. In Figure 8, we see that training with additional iterations improves performance and optimization accuracy. We stress that the exact form of this relationship depends on the $Q$-value estimator and other factors. We present additional results in Appendix B.6.

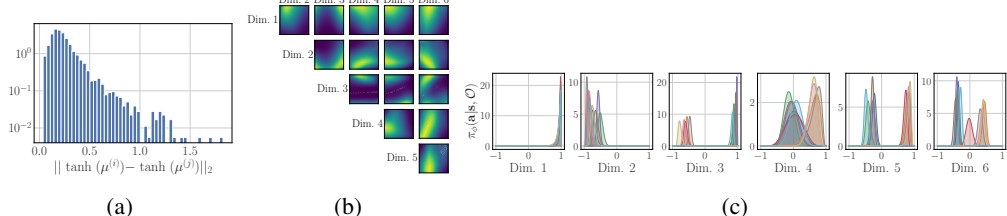

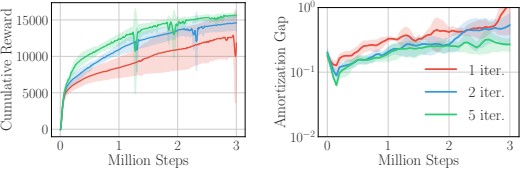

(a)              (b)                                  (c)

Figure 7: **Multiple Policy Modes**. **(a)** Histogram of distances between policy means ($\mu$) across optimization runs ($i$ and $j$) over seeds and states on `Walker2d-v2` at 3 million environment steps. For the state with the largest distance, **(b)** shows the projected optimization surface on each pair of action dimensions, and **(c)** shows the policy density for 10 optimization runs.

Figure 8: **Varying Iterations During Training**. Performance (**Left**) and estimated amortization gap (**Right**) for varying numbers of policy optimization iterations per step during training on `HalfCheetah-v2`. Increasing the iterations generally improves performance and decreases the estimated amortization gap.

Figure 9: **Zero-shot generalization** of iterative amortization from model-free (MF) to model-based (MB) value estimates.

### 4.2.5 Generalizing to Model-Based Value Estimates

Direct amortization is a purely feedforward process and is therefore incapable of generalizing to new objectives. In contrast, because iterative amortization is formulated through gradient-based feedback, such optimizers may be capable of generalizing to new objective estimators, as shown in Figure 1. To demonstrate this capability further, we apply iterative amortization with model-based value estimators, using a learned deterministic model on `HalfCheetah-v2` (see Appendix A.5). We evaluate the generalizing capabilities in Figure 9 by transferring the policy optimizer from a model-free agent to a model-based agent. Iterative amortization generalizes to these new value estimates, *instantly* recovering the performance of the model-based agent. This highlights the opportunity for instantly incorporating new tasks, goals, or model estimates into policy optimization.

## 5 Discussion

We have introduced iterative amortized policy optimization, a flexible and powerful policy optimization technique. In so doing, we have identified KL-regularized policy networks as a form of direct amortization, highlighting several limitations: 1) limited accuracy, as quantified by the amortization gap, 2) restriction to a single estimate, limiting exploration, and 3) inability to generalize to new objectives, limiting the transfer of these policy optimizers. As shown through our empirical analysis, iterative amortization provides a step toward improving each of these restrictions, with accompanying improvements in performance over direct amortization. Thus, iterative amortization can serve as a drop-in replacement and improvement over direct policy networks in deep RL.

This improvement, however, is accompanied by added challenges. As highlighted in Section 3.3, improved policy optimization can exacerbate issues in $Q$-value estimation stemming from positive bias. Note that this is not unique to iterative amortization, but applies broadly to any improved optimizer. We have provided a simple solution that involves adjusting a factor, $\beta$, to counteract this bias. Yet, we see this as an area for further investigation, perhaps drawing on insights from the offline RL community [49]. In addition to value estimation issues, iterative amortized policy optimization incurs computational costs that scale linearly with the number of iterations. This is in comparison with direct amortization, which has constant computational cost. Fortunately, unlike standard optimizers, iterative amortization adaptively tunes step sizes. Thus, relative improvements

rapidly diminish with each additional iteration, enabling accurate optimization with exceedingly few iterations. In practice, even a single iteration per time step can work surprisingly well.

Although we have discussed three separate limitations of direct amortization, these factors are highly interconnected. By broadening policy optimization to an iterative procedure, we automatically obtain a potentially more accurate and general policy optimizer, with the capability of obtaining multiple modes. While our analysis suggests that improved exploration resulting from multiple modes is the primary factor affecting performance, future work could tease out these effects further and assess the relative contributions of these improvements in additional environments. We are hopeful that iterative amortized policy optimization, by providing a more powerful, exploratory, and general optimizer, will enable a range of improved RL algorithms.

## Acknowledgments and Disclosure of Funding

JM acknowledges Scott Fujimoto for helpful discussions. This work was funded in part by NSF #1918839 and Beyond Limits. JM is currently employed by Google DeepMind. The authors declare no other competing interests related to this work.

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
