# A Experiment Details

Accompanying code is available at `github.com/joelouismarino/variational_rl`. Experiments were implemented in `PyTorch`[4] [61], analyzed with `NumPy`[5] [34], and visualized with `Matplotlib`[6] [39] and `seaborn`[7] [82]. Logging and experiment management was handled through `Comet` [18]. Experiments were performed on NVIDIA 1080Ti GPUs with Intel i7 8-core processors (@4.20GHz) on local machines, with each experiment requiring on the order of 2 days to 1 week to complete. We obtained the MuJoCo [78] software library through an Academic Lab license.

## A.1 2D Plots

In Figures 1 and 2, we plot the estimated variational objective, $\mathcal{J}$, as a function of two dimensions of the policy mean, $\mu$. To create these plots, we first perform policy optimization (direct amortization in Figure 1 and iterative amortization in Figure 2), estimating the policy mean and variance. This is performed using on-policy trajectories from evaluation episodes (for a direct agent in Figure 1 and an iterative agent in Figure 2). While holding all other dimensions of the policy constant, we then estimate the variational objective while varying two dimensions of the mean (1 & 3 in Figure 1 and 2 & 6 in Figure 2). Iterative amortization is additionally performed while preventing any updates to the constant dimensions. Even in this restricted setting, iterative amortization is capable of optimizing the policy. Additional 2D plots comparing direct vs. iterative amortization on other environments are shown in Figure 19, where we see similar trends.

## A.2 Value Bias Estimation

We estimate the bias in the $Q$-value estimator using a similar procedure as [25], comparing the estimate of the $Q$-networks ($\widehat{Q}_\pi$) with a Monte Carlo estimate of the future objective in the actual environment, $Q_\pi$, using a set of state-action pairs. To enable comparison across setups, we collect 100 state-action pairs using a uniform random policy, then evaluate the estimator's bias, $\mathbb{E}_{\mathbf{s},\mathbf{a}} \left[ \widehat{Q}_\pi - Q_\pi \right]$, throughout training. To obtain the Monte Carlo estimate of $Q_\pi$, we use 100 action samples, which are propagated through all future time steps. The result is discounted using the same discounting factor as used during training, $\gamma = 0.99$, as well as the same Lagrange multiplier, $\alpha$. Figure 3 shows the mean and $\pm$ standard deviation across the 100 state-action pairs.

## A.3 Amortization Gap Estimation

Calculating the amortization gap in the RL setting is challenging, as properly evaluating the variational objective, $\mathcal{J}$, involves unrolling the environment. During training, the objective is estimated using a set of $Q$-networks and/or a learned model. However, finding the optimal policy distribution, $\widehat{\pi}$, under these learned value estimates may not accurately reflect the amortization gap, as the value estimator likely contains positive bias (Figure 3). Because the value estimator is typically locally accurate near the current policy, we estimate the amortization gap by performing gradient ascent on $\mathcal{J}$ w.r.t. the policy distribution parameters, $\lambda$, initializing from the amortized estimate (from $\pi_\phi$). This is a form *semi-amortized* variational inference [37, 47, 43]. We use the Adam optimizer [45] with a learning rate of $5 \times 10^{-3}$ for 100 gradient steps, which we found consistently converged. This results in the estimated optimized $\widehat{\pi}$. We estimate the gap using 100 on-policy states, calculating $\mathcal{J}(\theta, \widehat{\pi}) - \mathcal{J}(\theta, \pi)$, i.e. the improvement in the objective after gradient-based optimization. Figure 6 shows the resulting mean and $\pm$ standard deviation. We also run iterative amortized policy optimization for an additional 5 iterations during this evaluation, empirically yielding an additional decrease in the estimated amortization gap.

---

[4] `https://github.com/pytorch/pytorch/blob/master/LICENSE`
[5] `https://numpy.org/doc/stable/license.html`
[6] `https://matplotlib.org/stable/users/license.html`
[7] `https://github.com/mwaskom/seaborn/blob/master/LICENSE`

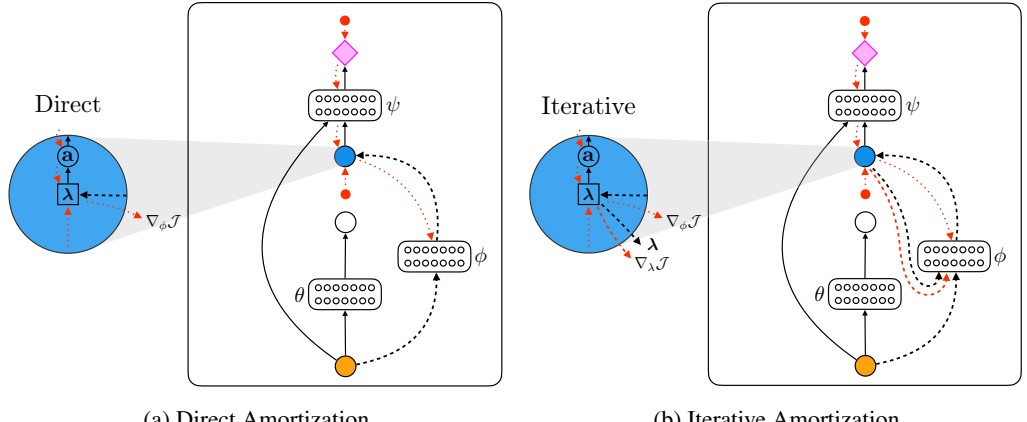

|         | Direct | Iterative |
|---------|:------:|:---------:|
| (a) Direct Amortization | | (b) Iterative Amortization |

Figure 10: **Amortized Optimizers**. Diagrams of **(a)** direct and **(b)** iterative amortized policy optimization. As in Figure 1, larger circles represent probability distributions, and smaller red circles represent terms in the objective. Red dotted arrows represent gradients. In addition to the state, $\mathbf{s}_t$, iterative amortization uses the current policy distribution estimate, $\boldsymbol{\lambda}$, and the policy optimization gradient, $\nabla_{\boldsymbol{\lambda}}\mathcal{J}$, to iteratively optimize $\mathcal{J}$. Like direct amortization, the optimizer network parameters, $\phi$, are updated using $\nabla_{\phi}\mathcal{J}$. This generally requires some form of stochastic gradient estimation to differentiate through $\mathbf{a}_t \sim \pi(\mathbf{a}_t|\mathbf{s}_t, \mathcal{O}; \boldsymbol{\lambda})$.

### A.4 Hyperparameters

Our setup follows that of soft actor-critic (SAC) [31, 32], using a uniform action prior, i.e. entropy regularization, and two $Q$-networks [25]. Off-policy training is performed using a replay buffer [53, 58]. Training hyperparameters are given in Table 7.

**Temperature** Following [32], we adjust the temperature, $\alpha$, to maintain a specified entropy constraint, $\epsilon_{\alpha} = |\mathcal{A}|$, where $|\mathcal{A}|$ is the size of the action space, i.e. the dimensionality.

Table 1: **Policy Inputs & Outputs**.

|           | Inputs | Outputs |
|-----------|:------:|:-------:|
| Direct    | $\mathbf{s}$ | $\boldsymbol{\lambda}$ |
| Iterative | $\mathbf{s}, \boldsymbol{\lambda}, \nabla_{\boldsymbol{\lambda}}\mathcal{J}$ | $\boldsymbol{\delta}, \boldsymbol{\omega}$ |

Table 2: **Policy Networks**.

| Hyperparameter | Value |
|----------------|:-----:|
| Number of Layers | 2 |
| Number of Units / Layer | 256 |
| Non-linearity | ReLU |

**Policy** We use the same network architecture (number of layers, units/layer, non-linearity) for both direct and iterative amortized policy optimizers (Table 2). Each policy network results in Gaussian distribution parameters, and we apply a `tanh` transform to ensure $\mathbf{a} \in [-1, 1]$ [31]. In the case of a Gaussian, the distribution parameters are $\boldsymbol{\lambda} = [\boldsymbol{\mu}, \boldsymbol{\sigma}]$. The inputs and outputs of each optimizer form are given in Table 1. Again, $\boldsymbol{\delta}$ and $\boldsymbol{\omega}$ are respectively the update and gate of the iterative amortized optimizer (Eq. 12 in the main text), each of which are defined for both $\boldsymbol{\mu}$ and $\boldsymbol{\sigma}$. Following [55], we apply layer normalization [9] individually to each of the inputs to iterative amortized optimizers. We initialize iterative amortization with $\boldsymbol{\mu} = \mathbf{0}$ and $\boldsymbol{\sigma} = \mathbf{1}$, however, these could be initialized from a learned action prior [54].

$Q$**-value** We investigated two $Q$-value network architectures. Architecture A (Table 3) is the same as that from [31]. Architecture B (Table 4) is a wider, deeper network with highway connectivity [72], layer normalization [9], and ELU nonlinearities [17]. We initially compared each $Q$-value network architecture using each policy optimizer on each environment, as shown in Figure 11. The results in Figure 5 were obtained using the better performing architecture in each case, given in Tables 5 & 6. As in [25], we use an ensemble of 2 separate $Q$-networks in each experiment.

| Table 3: $Q$-**value Network Architecture A**. | |
|---|---|
| Hyperparameter | Value |
| Number of Layers | 2 |
| Number of Units / Layer | 256 |
| Non-linearity | ReLU |
| Layer Normalization | False |
| Connectivity | Sequential |

| Table 4: $Q$-**value Network Architecture B**. | |
|---|---|
| Hyperparameter | Value |
| Number of Layers | 3 |
| Number of Units / Layer | 512 |
| Non-linearity | ELU |
| Layer Normalization | True |
| Connectivity | Highway |

Table 5: $Q$-**value Network Architecture by Environment**.

| | InvertedPendulum-v2 | InvertedDoublePendulum-v2 | Hopper-v2 | HalfCheetah-v2 | Walker2d-v2 | Ant-v2 |
|---|---|---|---|---|---|---|
| Direct | A | A | A | B | A | B |
| Iterative | A | A | A | A | B | B |

(a) `Hopper-v2`  (b) `HalfCheetah-v2`  (c) `Walker2d-v2`  (d) `Ant-v2`

Figure 11: **Value Architecture Comparison**. Plots show performance for $\geq 3$ seeds for each value architecture (A or B) for each policy optimization technique (direct or iterative). Note: results for iterative + B on `Hopper-v2` were obtained with an overly pessimistic value estimate ($\beta = 2.5$ rather than $\beta = 1.5$) and are consequently worse.

**Value Pessimism** ($\beta$)    As discussed in Section 3.2.2, the increased flexibility of iterative amortization allows it to potentially exploit inaccurate value estimates. We increased the pessimism hyperparameter, $\beta$, to further penalize variance in the value estimate. Experiments with direct amortization use the default $\beta = 1$ in all environments, as we did not find that increasing $\beta$ helped in this setup. For iterative amortization, we use $\beta = 1.5$ on `Hopper-v2` and $\beta = 2.5$ on all other environments. This is only applied during training; while collecting data in the environment, we use $\beta = 1$ to not overly penalize exploration.

### A.5    Model-Based Value Estimation

For model-based experiments, we use a single, deterministic model together with the ensemble of 2 $Q$-value networks (discussed above).

**Model**    We use separate networks to estimate the state transition dynamics, $p_{\text{env}}(\mathbf{s}_{t+1}|\mathbf{s}_t, \mathbf{a}_t)$, and reward function, $r(\mathbf{s}_t, \mathbf{a}_t)$. The network architecture is given in Table 8. Each network outputs the mean of a Gaussian distribution; the standard deviation is a separate, learnable parameter. The reward network directly outputs the mean estimate, whereas the state transition network outputs a residual estimate, $\Delta_{\mathbf{s}_t}$, yielding an updated mean estimate through:

$$\boldsymbol{\mu}_{\mathbf{s}_{t+1}} = \mathbf{s}_t + \Delta_{\mathbf{s}_t}.$$

**Model Training**    The state transition and reward networks are both trained using maximum log-likelihood training, using data examples from the replay buffer. Training is performed at the same frequency as policy and $Q$-network training, using the same batch size (256) and network optimizer. However, we perform $10^3$ updates at the beginning of training, using the initial random steps, in order to start with a reasonable model estimate.

**Value Estimation**    To estimate $Q$-values, we combine short model rollouts with the model-free estimates from the $Q$-networks. Specifically, we unroll the model and policy, obtaining state, reward,

Table 6: $Q$-**value Network Architecture by Environment (Continued)**.

|  | Reacher-v2 | Swimmer-v2 | Humanoid-v2 | HumanoidStandup-v2 |
|---|---|---|---|---|
| Direct | A | A | B | B |
| Iterative | A | A | B | B |

Table 7: **Training Hyperparameters**.

| Hyperparameter | Value |
|---|---|
| Discount Factor ($\gamma$) | 0.99 |
| $Q$-network Update Rate ($\tau$) | $5 \cdot 10^{-3}$ |
| Network Optimizer | Adam |
| Learning Rate | $3 \cdot 10^{-4}$ |
| Batch Size | 256 |
| Initial Random Steps | $5 \cdot 10^3$ |
| Replay Buffer Size | $10^6$ |

Table 8: **Model Network Architectures**.

| Hyperparameter | Value |
|---|---|
| Number of Layers | 2 |
| Number of Units / Layer | 256 |
| Non-linearity | Leaky ReLU |
| Layer Normalization | True |

Table 9: **Model-Based Hyperparameters**.

| Hyperparameter | Value |
|---|---|
| Rollout Horizon, $h$ | 2 |
| Retrace $\lambda$ | 0.9 |
| Pre-train Model Updates | $10^3$ |
| Model-Based Value Targets | True |

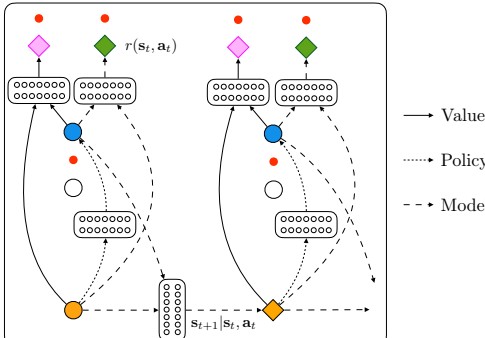

Figure 12: **Model-Based Value Estimation**. Diagram of model-based value estimation (shown with direct amortization). For clarity, the diagram is shown without the policy prior network, $p_\theta(\mathbf{a}_t|\mathbf{s}_t)$. The model consists of a deterministic reward estimate, $r(\mathbf{s}_t, \mathbf{a}_t)$, (green diamond) and a state estimate, $\mathbf{s}_{t+1}|\mathbf{s}_t, \mathbf{a}_t$, (orange diamond). The model is unrolled over a horizon, $H$, and the $Q$-value is estimated using the Retrace estimator [59], given in Eq. 14.

and policy estimates at current and future time steps. We then apply the $Q$-value networks to these future state-action estimates. Future rewards and value estimates are combined using the Retrace estimator [59]. Denoting the estimate from the $Q$-network as $\widehat{Q}_\psi(\mathbf{s}, \mathbf{a})$ and the reward estimate as $\widehat{r}(\mathbf{s}, \mathbf{a})$, we calculate the $Q$-value estimate at the current time step as

$$\widehat{Q}_\pi(\mathbf{s}_t, \mathbf{a}_t) = \widehat{Q}_\psi(\mathbf{s}_t, \mathbf{a}_t) + \mathbb{E}\left[\sum_{t'=t}^{t+h} \gamma^{t'-t} \lambda^{t'-t} \delta_{t'}\right], \tag{14}$$

where $\delta_{t'}$ is the estimated temporal difference:

$$\delta_{t'} \equiv \widehat{r}(\mathbf{s}_{t'}, \mathbf{a}_{t'}) + \gamma \widehat{V}_\psi(\mathbf{s}_{t'+1}) - \widehat{Q}_\psi(\mathbf{s}_{t'}, \mathbf{a}_{t'}), \tag{15}$$

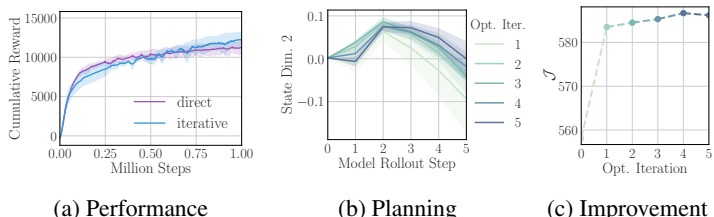

(a) Performance       (b) Planning       (c) Improvement

Figure 13: **Model-Based Value Estimates**. **(a)** Performance comparison of direct and iterative amortization using model-based value estimates. Curves show the mean and $\pm$ std. dev. over 4 random seeds. **(b)** Planned trajectories over policy optimization iterations. **(c)** The corresponding estimated objective increases over iterations.

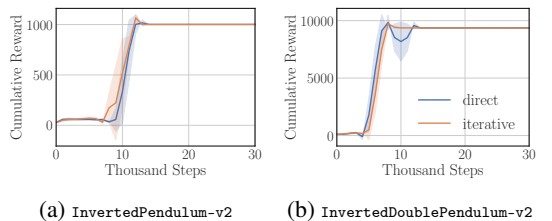

(a) `InvertedPendulum-v2`     (b) `InvertedDoublePendulum-v2`

Figure 14: **Pendulum Performance Comparison**. Curves show the mean and $\pm$ std. dev. over 5 random seeds.

Figure 15: **Amortization Gap** for direct + NF on `HalfCheetah-v2`.

$\lambda$ is an exponential weighting factor, $h$ is the rollout horizon, and the expectation is evaluated under the model and policy. In the variational RL setting, the state-value, $V_\pi(\mathbf{s})$, is

$$V_\pi(\mathbf{s}) = \mathbb{E}_\pi \left[ Q_\pi(\mathbf{s}, \mathbf{a}) - \alpha \log \frac{\pi(\mathbf{a}|\mathbf{s}, \mathcal{O})}{p_\theta(\mathbf{a}|\mathbf{s})} \right]. \tag{16}$$

In Eq. 14, we approximate $V_\pi$ using the $Q$-network to approximate $Q_\pi$ in Eq. 16, yielding $\widehat{V}_\psi(\mathbf{s})$. Finally, to ensure consistency between the model and the $Q$-value networks, we use the model-based estimate from Eq. 14 to provide target values for the $Q$-networks, as in [40].

**Future Policy Estimates** Evaluating the expectation in Eq. 14 requires estimates of $\pi$ at future time steps. This is straightforward with direct amortization, which employs a feedforward policy, however, with iterative amortization, this entails recursively applying an iterative optimization procedure. Alternatively, we could use the prior, $p_\theta(\mathbf{a}|\mathbf{s})$, at future time steps, but this does not apply in the max-entropy setting, where the prior is uniform. For computational efficiency, we instead learn a separate direct (amortized) policy for model-based rollouts. That is, with iterative amortization, we create a separate direct network using the same hyperparameters from Table 2. This network distills iterative amortization into a direct amortized optimizer, through the KL divergence, $D_{\text{KL}}(\pi_{\text{it.}}||\pi_{\text{dir.}})$. Rollout policy networks are common in model-based RL [69, 62].

**Model-Based Results** In Figure 13, we show results using iterative amortization with model-based value estimates. Iterative amortization offers a slight improvement over direct amortization in terms of performance at 1 million steps. As in the model-free case, iterative amortization results in improvement over iterations (Fig. 13c), but now as a result of planning trajectories (Fig. 13b).

# B   Additional Results

## B.1   Pendulum Environments

In Figure 14, we present additional results on the remaining two MuJoCo environments from OpenAI gym: `InvertedPendulum-v2` and `InvertedDoublePendulum-v2`. The curves show the mean and standard deviation over 5 random seeds. Iterative amortization performs comparably with direct amortization on these simpler environments.

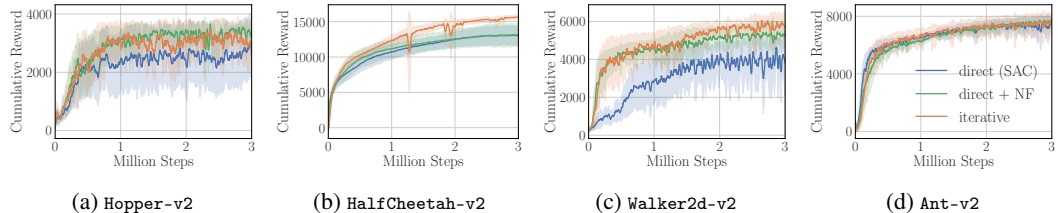

(a) `Hopper-v2`  (b) `HalfCheetah-v2`  (c) `Walker2d-v2`  (d) `Ant-v2`

Figure 16: **Performance Comparison with NF Policies**. Curves show the mean $\pm$ std. dev. of performance over 5 random seeds.

## B.2  Comparison with Normalizing Flow-Based Policies

Iterative amortization is capable of estimating multiple policy modes, potentially yielding improved exploration. Thus, the benefits of iterative amortization may come purely from this effective improvement in the policy distribution. To test this hypothesis, we compare with direct amortization with normalizing flow-based (NF) policies, formed using two inverse autoregressive flow transforms [46]. Each transform is parameterized by a network with 2 layers of 256 units with `ReLU` activation, and we reverse the action dimension ordering between the transforms to model dependencies in both directions. In Figure 16, we plot performance on a subset of environments, where we see that direct + NF closes the performance gap on `Hopper-v2` and `Walker2d-v2` but is unable to close the gap on `HalfCheetah-v2`. In Figure 15, we see that direct + NF is also unable to close the amortization gap early on during training. Thus, improved optimization of iterative amortization, rather than purely improved exploration, does appear to play some role in the performance improvements.

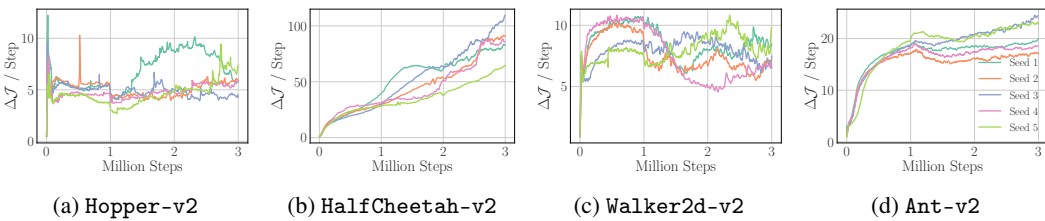

(a) `Hopper-v2`  (b) `HalfCheetah-v2`  (c) `Walker2d-v2`  (d) `Ant-v2`

Figure 17: **Per-Step Improvement**. Each plot shows the per-step improvement in the estimated variational RL objective, $\mathcal{J}$, throughout training resulting from iterative amortized policy optimization. Each curve denotes a different random seed.

## B.3  Improvement per Step

In Figure 17, we plot the average improvement in the variational objective per step throughout training, with each curve showing a different random seed. That is, each plot shows the average change in the variational objective after running 5 iterations of iterative amortized policy optimization. With the exception of `HalfCheetah-v2`, the improvement remains relatively constant throughout training and consistent across seeds.

## B.4  Comparison with Iterative Optimizers

Iterative amortized policy optimization obtains the *accuracy* benefits of iterative optimization while retaining the *efficiency* benefits of amortization. In Section 4, we compared the accuracy of iterative and direct amortization, seeing that iterative amortization yields reduced amortization gaps (Figure 6) and improved performance (Figure 5). In this section, we compare iterative amortization with two popular iterative optimizers: Adam [45], a gradient-based optimizer, and cross-entropy method (CEM) [66], a gradient-free optimizer.

To compare the accuracy and efficiency of the optimizers, we collect 100 states for each seed in each environment from the model-free experiments in Section 4.2.2. For each optimizer, we optimize the variational objective, $\mathcal{J}$, starting from the same initialization. Tuning the step size, we found that 0.01 yielded the steepest improvement without diverging for both Adam and CEM. Gradients are

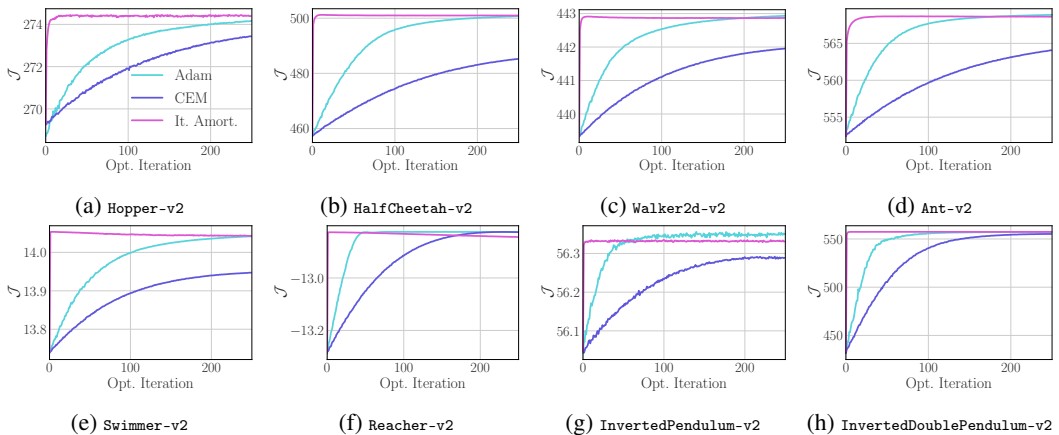

(a) `Hopper-v2`  (b) `HalfCheetah-v2`  (c) `Walker2d-v2`  (d) `Ant-v2`

(e) `Swimmer-v2`  (f) `Reacher-v2`  (g) `InvertedPendulum-v2`  (h) `InvertedDoublePendulum-v2`

Figure 18: **Comparison with Iterative Optimizers**. Average estimated objective over policy optimization iterations, comparing with Adam [45] and CEM [66]. These iterative optimizers require over an order of magnitude more iterations to reach comparable performance with iterative amortization, making them impractical in many applications.

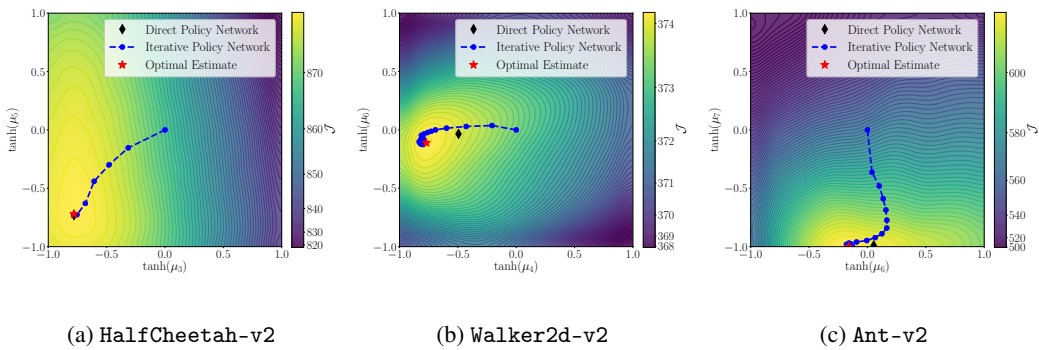

(a) `HalfCheetah-v2`  (b) `Walker2d-v2`  (c) `Ant-v2`

Figure 19: **2D Optimization Plots**. Each plot shows the optimization objective over two dimensions of the policy mean, $\mu$. This optimization surface contains the value function trained using a direct amortized policy. The black diamond, denoting the estimate of this direct policy, is generally near-optimal, but does not match the optimal estimate (red star). Iterative amortized optimizers are capable of generalizing to these surfaces in each case, reaching optimal policy estimates.

evaluated with 10 action samples. For CEM, we sample 100 actions and fit a Gaussian mean and variance to the top 10 samples. This is comparable with QT-Opt [41], which draws 64 samples and retains the top 6 samples.

The results, averaged across states and random seeds, are shown in Figure 18. CEM (gradient-free) is less efficient than Adam (gradient-based), which is unsurprising, especially considering that Adam effectively approximates higher-order curvature through momentum terms. However, Adam and CEM both require over *an order of magnitude* more iterations to reach comparable performance with iterative amortization. While iterative amortized policy optimization does not always obtain asymptotically optimal estimates, we note that these networks were trained with only 5 iterations, yet continue to improve and remain stable far beyond this limit. Finally, comparing wall clock time for each optimizer, iterative amortization is only roughly $1.25\times$ slower than CEM and $1.15\times$ slower than Adam, making iterative amortization still substantially more efficient.

## B.5  Additional 2D Optimization Plots

In Figure 1, we provided an example of suboptimal optimization resulting from direct amortization on the `Hopper-v2` environment. We also demonstrated that iterative amortization is capable of

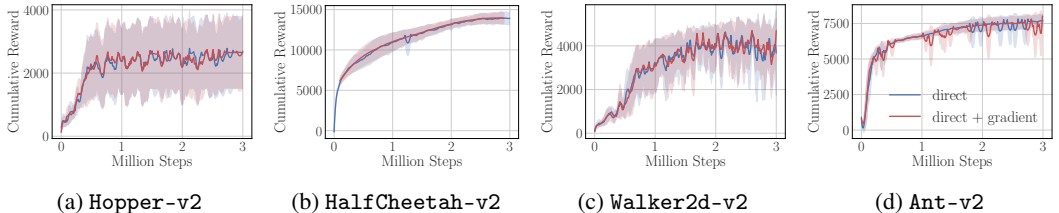

| (a) Hopper-v2 | (b) HalfCheetah-v2 | (c) Walker2d-v2 | (d) Ant-v2 |

Figure 20: **Test-Time Gradient-Based Optimization**. Each plot compares the performance of direct amortization vs. direct amortization with 50 additional gradient-based policy optimization iterations. Note that this additional optimization is only performed at test time.

automatically generalizing to this optimization surface, outperforming the direct amortized policy. To show that this is a general phenomenon, in Figure 19, we present examples of corresponding 2D plots for each of the other environments considered in this paper. As before, we see that direct amortization is near-optimal, but, in some cases, does not match the optimal estimate. In contrast, iterative amortization is able to find the optimal estimate, again, generalizing to the unseen optimization surfaces.

### B.6    Additional Optimization & the Amortization Gap

In Section 4, we compared the performance of direct and iterative amortization, as well as their estimated amortization gaps. In this section, we provide additional results analyzing the relationship between policy optimization and the performance in the actual environment. As emphasized in the main paper, this relationship is complex, as optimizing an inaccurate $Q$-value estimate does not improve task performance. Likewise, optimization improvement accrues over the course of training, facilitating collecting high-value state-action pairs in the environment.

The amortization gap quantifies the suboptimality in the objective, $\mathcal{J}$, of the policy estimate. As described in Section A.3, we estimate the optimized policy by performing additional gradient-based optimization on the policy distribution parameters (mean and variance). However, as noted, this gap is relatively small compared to the objective itself. Thus, when we deploy this optimized policy for evaluation in the actual environment, as shown for direct amortization in Figure 20, we do not observe a noticeable difference in performance.

Likewise, in Section 4.2.2, we observed that using additional amortized iterations during evaluation further decreased the amortization gap for iterative amortization. Yet, when we deploy this more fully optimized policy in the environment, as shown in Figure 21, we do not generally observe a corresponding performance improvement. In fact, on `HalfCheetah-v2` and `Walker2d-v2`, we observe a slight *decrease* in performance. This further highlights the fact that additional policy optimization may exploit inaccurate $Q$-value estimates.

However, importantly, in Figures 20 and 21, the additional policy optimization is only performed for evaluation. That is, the data collected with the more fully optimized policy is not used for training and therefore cannot be used to correct the inaccurate value estimates. Thus, while more accurate policy optimization, as quantified by the amortization gap, may not substantially affect *evaluation* performance, it can play a role in improving *training*.

This aspect is explored in Figures 22 & 23, where we plot the performance and amortization gap of iterative amortization with varying numbers of iterations (1, 2 and 5) throughout training. While the trend is not exact in all cases, we generally observe that increasing iterations yields improved performance and more accurate optimization. `Walker2d-v2` provides an interesting example. Even with a single iteration, we see that iterative amortization outperforms direct amortization, suggesting that multi-modality is the dominant factor for improved performance here. Yet, 1 iteration is slightly worse compared with 2 and 5 iterations early in training, both in terms of performance and optimization. As the amortization gap decreases later in training, we see that the performance gap ultimately decreases. Further work could help to analyze this process in even more detail.

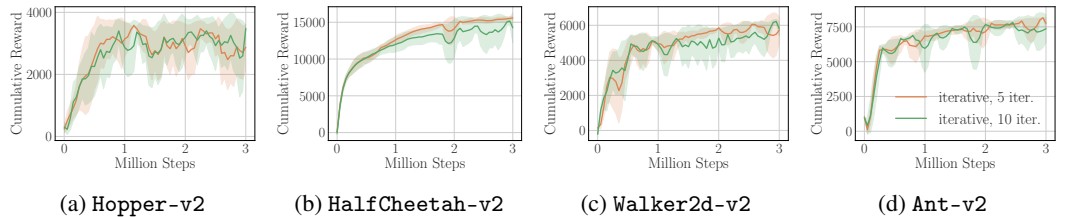

Figure 21: **Additional Amortized Test-Time Iterations**. Each plot compares the performance of iterative amortization (trained with 5 iterations) vs. the same agent with an additional 5 iterations at evaluation. Performance remains similar or slightly worse.

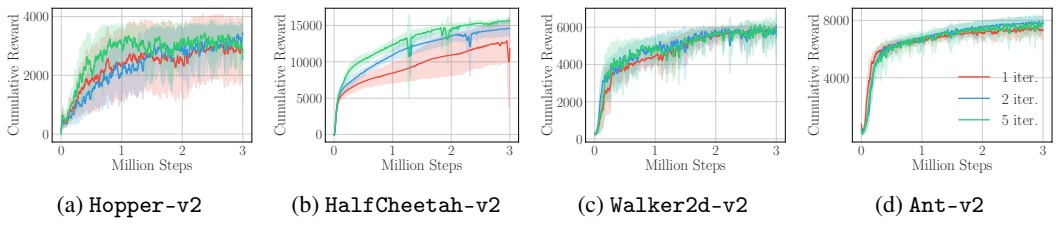

Figure 22: **Performance of Varying Iterations During Training**.

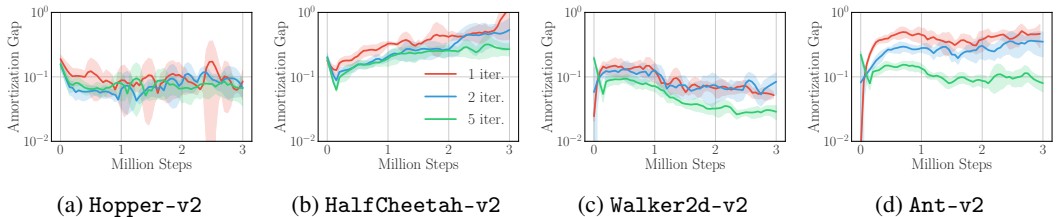

Figure 23: **Amortization Gap of Varying Iterations During Training**.

## B.7   Multiple Policy Estimates

As discussed in Section 3.2, iterative amortization has the added benefit of potentially obtaining multiple policy distribution estimates, due to stochasticity in the optimization procedure as well as initialization. In contrast, unless latent variables are incorporated into the policy, direct amortization is limited to a single policy estimate. In Section 4.3.2, we analyzed multi-modality by comparing the distance between difference optimization runs of iterative amortization on `Walker2d-v2`. Here, we present the same analysis on each of the other three environments considered in this paper.

Again, we perform 10 separate runs of policy optimization per state and evaluate the L2 distance between the means of these policy estimates after applying the `tanh` transform. Note that in MuJoCo action spaces, which are bounded to $[-1, 1]$, the maximum distance is $2\sqrt{|\mathcal{A}|}$, where $|\mathcal{A}|$ is the size of the action space. We evaluate the policy mean distance over 100 states and all 5 experiment seeds. Results are shown in Figures 25, 24, and 26, where we plot **(a)** the histogram of distances between policy means ($\boldsymbol{\mu}$) across optimization runs ($i$ and $j$) over seeds and states at 3 million environment steps, **(b)** the projected optimization surface on each pair of action dimensions for the state with the largest distance, and **(c)** the policy density for 10 optimization runs. As before, we see that some subset of states retain multiple policy modes.

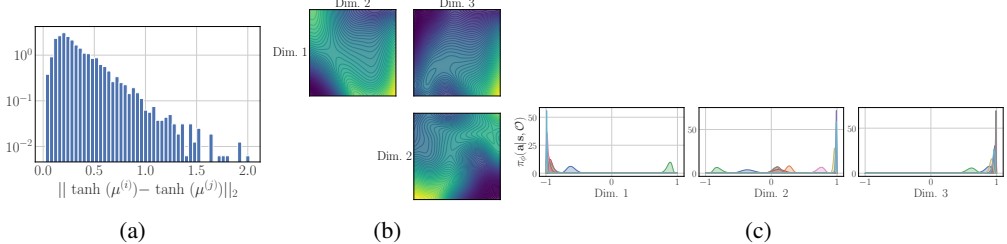

(a)             (b)             (c)

Figure 24: **Multiple Policy Modes on** `Hopper-v2`.

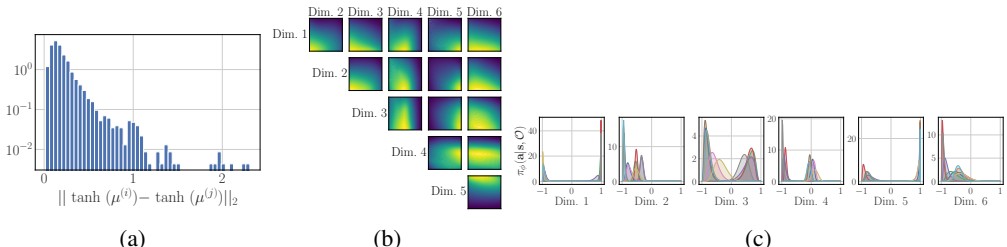

(a)             (b)             (c)

Figure 25: **Multiple Policy Modes on** `HalfCheetah-v2`.

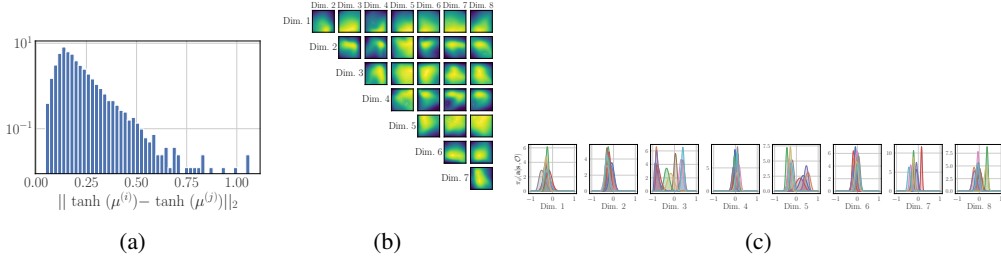

(a)             (b)             (c)

Figure 26: **Multiple Policy Modes on** `Ant-v2`.