# OpenReview forum: "Iterative Amortized Policy Optimization"
_NeurIPS.cc/2021/Conference — NeurIPS 2021 Poster_

### Official Review · Reviewer_CT92 · 2021-07-09

**Rating:** 7
**Confidence:** 4

**Summary:**

This paper introduces a novel approach to policy optimization which leverages the connection between amortized variational inference and policy-based RL with function approximation. Specifically, rather than rely on a "direct" parameterization of the state-conditioned distribution over actions, in which an input state is mapped directly to distribution parameters (e.g., mean and variance), the approach adds an additional optimization procedure over the distribution parameters themselves, rather than those of the policy network. Experimental results are then used to demonstrate the benefits of this approach for performance, accuracy, and flexibility in the optimization process.

**Limitations And Societal Impact:**

See the main review for a discussion of limitations and suggestions for improvement. I do not believe there are obvious societal impacts for this work.

**Main Review:**

Originality: The application of iterative amortization to policy optimization for RL is interesting and novel to the best of my knowledge. The connections between variational inference and RL with function approximation are deep yet not fully explored, and this is an excellent example of the interesting insights that can be gained through the lens of this relationship. The relationship to previous work and associated citations appear thorough.

Quality: The submission appears technically sound, and the claims seem to be supported. However, it would be helpful if the authors discussed the tradeoffs inherent in iterative amortization (e.g., how is performance improvement balanced by increased computational expense?). Additionally, iterative amortization appears to help on some tasks and not at all on others. What properties of an environment make iterative amortization more or less useful? How do these properties vary among the tasks used in the paper? These are important questions from both a practical and theoretical standpoint. The analysis of the contribution of multimodality to performance is likely a good first step in this direction. There are also a number of supporting results in the appendix. Minor: For eq. 2, should there be a $\rho(s_1)$ prefixed to the righthand side of the equation? And shouldn't the upper limit of the products in both eqs. 1 and 2 be $T-1$?

Clarity: The paper is mostly clear and well-organized. The background section and discussion of value overestimation were particularly enjoyable to read. Minor: Figure 7b is difficult to parse--I'm unclear as to what takeaway is intended. It also would be helpful to mark the exact section of the appendix in which supplementary results appear.

Significance: The magnitude of the performance gain on the majority of Mujoco tasks is not significant. It's difficult to gauge the significance of reducing the amortization gap--why this gap appear to grow over the course of training (Figure 8, right)?  While the raw performance improvement due to the proposed approach is not generally notable, I think the main significance of this work is captured by results like the ability to flexibly adapt to new objectives, such as to model-based value estimation. Namely, its significance lies in its novelty and the potential for follow-up work. Some theoretical analysis would be helpful in developing a deeper understanding.



Given the above factors, I recommend acceptance.

**Time Spent Reviewing:**

7 hours

---

> ### Author Response · Authors · 2021-08-06
> **Response to Reviewer CT92**
>
> Thank you for your review. We are glad that you found the paper novel, thorough, and technically sound. To address your questions:
>
> We will include an additional discussion on the computational/performance trade-offs of iterative amortization. Briefly, computational cost scales linearly with the number of iterations per environment step, whereas there are diminishing performance improvements for additional iterations (see Figure B.9). However, even with a single iteration per step, one can obtain the other benefits of iterative amortization: with a single iteration, iterative amortization outperforms direct amortization on Walker2d.
>
> It is not entirely clear what properties of the environment and Q-value estimator are necessary for iterative amortization to yield significant improvements, and we agree that this would be a useful direction for further investigation. We suspect that there are multiple factors at play, including the complexity of the environment as well as the flexibility and accuracy of the Q-value estimator. We are glad that you appreciate our analysis on Walker2d as a meaningful first step in addressing this point, and we are hopeful that introducing the terminology of amortization to RL will inspire future works in this direction.
>
> You are correct on Eqs. 1 and 2. We will fix these in an updated version of the paper. Thank you for pointing this out.
>
> Regarding Figure 7b, the main takeaway is to note the multi-modal policy optimization surface, resulting in the multi-modal policy in Figure 7c. We will clarify this point in the main text.
>
> In the final version of the paper, we will provide more specific links to the supplementary material throughout the paper.
>
> As noted in the paper, the relationship between the amortization gap and performance varies across environments. However, we empirically demonstrate that improvements in the amortization gap are consistently accompanied by similar or improved performance. That is, there may be limits beyond which improvements in optimization do not yield improvements in performance. Regarding Figure 8 specifically, the cumulative reward on HalfCheetah is significantly larger than those on most other environments and tends to increase more throughout the course of training. We suspect that the increasing amortization gap (which is measured in absolute terms) may reflect the increasing magnitude of the Q-value estimates. Note that the amortization gap generally remains constant or decreases on the other environments, where the cumulative reward magnitude does not increase as much throughout training.
>
> Finally, we would like to thank you for noting the novelty of this perspective and the potential for future work afforded by our paper. We appreciate that you found these aspects appealing.

---

### Official Review · Reviewer_Embw · 2021-07-15

**Rating:** 7
**Confidence:** 4

**Summary:**

The authors, via a standard reinterpretation of reinforcement learning as an inference problem, discusses the resulting “amortization gap” that results from using policy networks to optimize the variational lower bound. In order to close this gap, the authors propose to train an iterative procedure parameterized by $\phi$, which uses estimated gradients of the return to iteratively refine $\pi(\cdot | s_t)$ for the current state $s_t$. They then provide an extensive experimental evaluation demonstrating their proposed method and several desirable properties, including adaptability to gradients estimated by different methods.

**Limitations And Societal Impact:**

The authors have adequately addressed the limitations of their approach. I particularly appreciated that the authors spent time addressing the value overestimation problem that is inevitably exacerbated by having a more expressive policy class.

**Main Review:**

**Originality:**
The authors leverage a known connection between variational inference and reinforcement learning to motivate their problem (which is known in the VI literature). Although the techniques are not novel, they are novel in their application to the RL setting.

**Quality:**
The submission is technically sound, with authors leveraging known techniques from the generative modeling literature. Furthermore, the authors present a very thorough experimental evaluation of their claims, with very interesting results. Although the performance gains in some of the tasks are not substantial, the demonstrated reduction in the amortization gap, inherent multi-modal behavior, and transferability between different value estimation methods are worthy of note.

As an additional experiment, it would be interesting to see whether the learned policy is robust to slight changes in the environment (e.g. changing weight/friction parameters of the MuJoCo simulation) as an extension of the generalization experiment.

**Clarity:**
The paper is mostly clear, although as someone who is not as familiar with the generative modeling literature, a few parts of the paper were a bit confusing from my perspective.
For example, in the context of RL, what is estimation referring to in lines 112-116 (I assume in this case that it is the process of choosing an action distribution)? I think it would be really helpful to have these connections explicitly mentioned.

**Significance:**
The paper introduces really interesting ideas, including having a trained policy which includes a trained optimizer that improves upon an initial action based on predicted returns. The work suggests several venues for future work, including across-task generalization, better policies for existing algorithms, among others.



**Time Spent Reviewing:**

5

---

> ### Author Response · Authors · 2021-08-06
> **Response to Reviewer Embw**
>
> Response:
>
> Thank you for your review. We are glad you found the paper novel, technically sound, and thorough. Regarding your questions:
>
> We agree that generalizing across environments is an interesting direction for investigation. We note that this still would require training a value estimator for this new environment. Alternatively, with a differentiable model or simulator, one could potentially directly adjust these parameters. We intend to investigate this setting and include the analysis in the final version of the paper. Thank you for the suggestion.
>
> Regarding lines 112-116, yes, you are correct. Estimation in this context refers to estimating the action distribution, i.e., policy, for a given state. We will clear up this ambiguity (and others) in the paper.

---

### Official Review · Reviewer_Cxw1 · 2021-07-16

**Rating:** 8
**Confidence:** 3

**Summary:**

This work analyzes policy optimization from an inference perspective. It first shows that policy optimization with entropy or KL regularization in DRL with a policy network is a form of amortized optimization where instead of directly optimizing the action distribution parameters, we optimize a network which outputs the action distribution parameters. This results in an amortization gap which negatively impacts performance and exploration. Next the authors propose an alternative iterative amortization method for training the policy network. This new approach is shown to have various benefits which the authors demonstrated through a series of experiments on the MuJoCo environments.

**Limitations And Societal Impact:**

Yes

**Main Review:**

Overall, I found this paper to be very insightful. It brings forward many interesting ideas which helps to bridge the gap between inference and RL. The paper was very easy to read. The authors did a good job of explaining the issues of direct amortization and presenting their novel technique. The diagrams were also very helpful! The discussion on amortization gap is particularly interesting and to the best of my knowledge not something that has previously been explored by the RL community. The experiments in this paper are very well designed in explaining the iterative amortization approach. I have two questions for the authors which I hope they can clarify in their response:

- In the section on value overestimation (Section 3.3), you mentioned that “If the policy can exploit regions of high uncertainty, the resulting target values will introduce positive bias into the estimate. More flexible policy optimizers exacerbate the problem, exploiting this uncertainty to a greater degree.” Could you elaborate on this claim a little further? Why are more powerful policies more prone to overestimation of the Q function? This seems to be a fairly central point to Section 3.3.
- Empirical performance of the iterative amortization approach does not seem to be that strong. While there are some improvements on some of the simpler environments. Performance on the most challenging MuJoCo environments (Ant, Humanoid, HumanoidStandup) appear to be on par with the direct amortization method. Any sense why the iterative approach does not result in any improvement on these environments?


**Time Spent Reviewing:**

4-5

---

> ### Author Response · Authors · 2021-08-06
> **Response to Reviewer Cxw1**
>
> Thank you for your review. We are glad you found the paper insightful and easy to read. To answer your questions:
>
> 1. Value overestimation results from bootstrapping the target Q-value estimate, which will have some regions with positive bias errors. Because actor-critic methods use a policy to maximize the target Q-value (i.e., find higher values), such methods will tend to exploit regions with positive bias. During the course of training, this results in a build-up of positive bias, leading to overestimated values overall. That is, due to the value maximization during policy optimization, any small (positive) errors in value estimates will become magnified. Because iterative amortization is a more powerful optimizer, it is more capable of exploiting positively biased target Q-value estimates. In other words, if you are able to find higher-value policy estimates (even if they are inaccurate), you are more prone to overestimate values overall. We see this as an issue to be addressed in value estimation, e.g., through better uncertainty estimates, rather than handicapping the policy optimizer. Accordingly, we use a simple uncertainty down-weighting scheme to mitigate value overestimation.
>
> 2. We agree that the performance differences are relatively modest on some environments. However, considering that we are only swapping out the policy optimizer, we believe that it is still surprising that we can consistently match or outperform direct policies. Direct policy networks are a central aspect of modern actor-critic RL algorithms, yet, we show that it is possible to still improve beyond this scheme. Regarding performance on the more challenging MuJoCo environments, we speculate that the Q-value networks may be a limiting factor here. The quality of the Q-value estimates sets an upper limit on performance, even with a perfect policy optimizer. Thus, direct and iterative policies may perform similarly for a relatively limited Q-value estimator (although we do see a slight performance boost on HumanoidStandup). Our paper introduces iterative amortized policy optimization, identifying several issues with current direct policies. Applying and analyzing this technique in the context of state-of-the-art value estimators, e.g., combinations of model-based and mode-free [Amos et al., 2020], would be an interesting direction for follow-up work.
>
> Brandon Amos, Samuel Stanton, Denis Yarats, and Andrew Gordon Wilson. On the model-based stochastic value gradient for continuous reinforcement learning. arXiv preprint arXiv:2008.12775, 2020.

---

> > ### Comment · Reviewer_Cxw1 · 2021-08-25
> > **Post-rebuttal response**
> >
> > I want to thank the authors for addressing my concerns. I am satisfied with the authors' response and I am keeping my original score. Good work!

---

### Official Review · Reviewer_fdJ1 · 2021-07-22

**Rating:** 6
**Confidence:** 2

**Summary:**

This paper proposes iterative amortized policy optimization, a class of methods that use the gradients online to optimize the policy iteratively.

**Limitations And Societal Impact:**

See the main review.

**Main Review:**

This paper proposes iterative amortized policy optimization, a class of methods that use the gradients online to optimize the policy iteratively. To the best of my knowledge, the idea is novel and original. The paper is generally well-written and easy to follow and in particular, I appreciate the authors making a detailed, aligned, and symmetric comparison of direct amortization and iterative amortization (the one-to-one pros and cons, as well as the graphical demonstration, and the algorithm, etc...). The empirical experiments are quite comprehensive, which includes quite some qualitative analysis of the claimed properties. And indeed, the proposed method is validated by the learning curves in MuJoCo continuous control tasks benchmark, as the proposed method outperforms and matches direct amortization in most continuous control tasks. Some questions and/or suggestions: 1. could you talk more about what's the added complexity of the proposed method? 2. In Fig.5 (c), while direct amortization seems to be very stable, iterative amortization seems to have two moments with sudden performance collapse, is there any reason why? 3. the increase on performance gains on MuJoCo tasks (Fig.5) and the corresponding decrease in amortization gap (Fig.6) do not seem to be be consistent, e.g. there is a significant decrease in amortization gap on Ant, while the expected reward is almost the same. So maybe it's good to investigate what brings the performance gains? 4. for exploration, it's also helpful to test the proposed algorithm in MuJoCo tasks with sparse rewards.




**Time Spent Reviewing:**

10

---

> ### Author Response · Authors · 2021-08-06
> **Response to Reviewer fdJ1**
>
> Thank you for your review. We are glad you found the paper novel, well-written, and comprehensive. We hope to answer your remaining questions here:
>
> 1. Regarding the computational complexity of iterative amortization, it scales linearly with the number of policy optimization iterations per-step. Each iteration requires a backward gradient pass through the Q-network or model, as well as a forward pass through the amortized policy network. However, by using a learned policy optimizer, even a single iteration can perform reasonably well. As shown in our experiments, there are diminishing benefits for additional iterations, and, in principle, one could adaptively tune the number of iterations. Regarding implementation complexity, we apply iterative amortized optimizers out-of-the-box, using the exact same policy network architectures as SAC. The only added challenge is increased value overestimation, which we address with a simple solution in the paper.
>
> 2. Across environments, iterative amortization does not appear to be significantly more or less stable than direct amortization. With the specific example of HalfCheetah, it is unclear why iterative amortization experiences some brief dips in performance. We speculate that this may be due to the more exploratory nature of iterative amortization, which is capable of jumping between policies. With an inaccurate or uncertain value estimate, this may result in more dips in performance. However, it is worth noting that iterative amortization results in higher performance and a lower variance across seeds. This suggests that, although direct amortization may be slightly more stable, it is getting stuck in various sub-optimal solutions.
>
> 3. We found that a decrease in the amortization gap consistently corresponds to an increase in performance (see Figure B.10 in the supplementary), however, as you note, the relationship between these changes is not the same across environments. This could be for a number of factors, including the accuracy of the Q-value estimate or intrinsic aspects of the environment dynamics and reward function. In our paper, we have brought the terminology of the amortization gap to the reinforcement learning community, along with substantial analysis. We agree, though, that a more in-depth investigation will be a useful direction for future work.
>
> 4. Our analysis has provided clear qualitative evidence for improvements in exploration. We agree that experiments on sparse-reward environments would be a useful addition, and we intend to include these in the final version of the paper. Thank you for this suggestion. We note, however, that improving policy optimization is only one aspect of exploration, and additionally improving Q-value uncertainty estimates or state-entropy maximization would provide a more comprehensive approach.
>
> Again, thank you for your review, and we hope that you will consider updating your score to reflect our response.

---

### Decision · Program_Chairs · 2021-09-27

**Decision:**

Accept (Poster)

**Comment:**

All the reviewers think introduce the advanced variational inference for making the policy flexible by exploiting the connection between RL and variational inference is interesting.

Although I recommend the acceptance for this submission based on the reviewers feedback, there are several issues should be extensively discussed:

- EBM closed-form for policy parametrization: in fact, with entropy-regularization, the policy will lie in energy-based model. Therefore, a natural competitor is advanced sampling algorithms, e.g., Langevin, HMC, SVGD [1], Wasserstein flow [2] etc, based on learned EBM.

- Performance tradeoff in computation, sample complexity, vs flexibility:  as the complicated parametrization introduced, the computation cost and sample complexity will increase. This should be carefully discussed, as almost all reviewers pointed out. From the empirical study, it seems the flexible policy family does not provide significant benefits, which diminishes the significance of this paper.

In sum, I think this paper is interesting and could be better if the authors carefully address these questions.

[1] Haarnoja, Tuomas, Haoran Tang, Pieter Abbeel, and Sergey Levine. "Reinforcement learning with deep energy-based policies." In International Conference on Machine Learning, pp. 1352-1361. PMLR, 2017.
[2] Zhang, Ruiyi, Changyou Chen, Chunyuan Li, and Lawrence Carin. "Policy optimization as wasserstein gradient flows." In International Conference on Machine Learning, pp. 5737-5746. PMLR, 2018.